# Phenotypic Spectrum of *KATNIP*-Associated Joubert Syndrome: Possible Association with Esophageal Atresia and Review of the Literature

**DOI:** 10.3390/genes16050524

**Published:** 2025-04-29

**Authors:** Maria Giovanna Tedesco, Ilaria Donati, Chiara Romeo, Sara Dal Bo, Chiara Nardini, Anna Maria Innoceta, Giulia Parmeggiani, Anna Patanè, Claudio Graziano

**Affiliations:** 1Unit of Medical Genetics, Department Maternal-Infantile, S. Maria Della Misericordia Hospital, 06129 Perugia, Italy; 2Medical Genetics Unit, MeLabeT Department, AUSL Romagna, 47521 Cesena, Italy; 3Diagnostic Neuroradiology, Department of Neuroscience, AUSL Romagna, 48121 Ravenna, Italy; 4Department of Pediatrics, Santa Maria Delle Croci Hospital, AUSL Romagna, 48121 Ravenna, Italy; 5R&I Genetics SRL, 35127 Padua, Italy

**Keywords:** *KATNIP*, Joubert syndrome, molar tooth sign, esophageal atresia

## Abstract

**Background:** Joubert syndrome (JS) is a multi-systemic ciliopathy, characterized by intellectual disability and congenital anomalies involving the brain, kidney, heart, and eye. Even if clinical presentation is variable, most authors consider a brain abnormality known as the molar tooth sign (MTS) as mandatory for diagnosis. About 40 genes were identified to be associated with JS, usually with an autosomal recessive pattern. *KATNIP* variants represent a rare cause of JS; only six families were previously reported. **Methods:** We performed exome sequencing in a child with a syndromic phenotype, described the clinical features and molecular findings, and performed a review of the literature to identify known individuals with pathogenic variants in *KATNIP*, highlighting clinical characteristics and gene-phenotype correlations. **Results:** Using exome sequencing, we identified a homozygous novel frameshift variant c.808del, p.Ser270ValfsTer28 in *KATNIP* in a 5-year-old male from a consanguineous family of Roma ethnic background. Notable clinical features of the proband include severe developmental delay, hypotonia, and post-axial polydactyly. He did not have MTS, but showed severe anemia and esophageal atresia, which was already reported in association with a *KATNIP* variant. We collected the phenotypes of all reported patients and discussed common and distinct features with respect to typical JS. Affected individuals shared JS clinical features, although the typical MTS was not always present, polydactyly and renal abnormalities were absent, while pituitary abnormalities were common. **Conclusions:** Our report provides new data for *KATNIP*-related JS, expanding the clinical phenotypic spectrum and suggesting a possible role of *KATNIP* defects in the development of esophageal atresia.

## 1. Introduction

Joubert syndrome (JS) is a multi-systemic ciliopathy that manifests early in life with hypotonia, developmental delay, abnormal eye movements (nystagmus, oculomotor apraxia), and respiratory pattern defects (apnea or tachypnea that may alternate). It is diagnosed by the presence of a peculiar cerebellar and brainstem malformation, known as the “molar tooth sign” (MTS) [1,2,3,4]. Intellectual disability is present in most, although a minority of individuals have normal cognition, and later involvement of other organs, such as the retina (chorioretinal coloboma, retinal dystrophy), kidney (cystic dysplasia evolving to nephronophthisis), liver (hepatic fibrosis), and skeleton (scoliosis, polydactyly), is frequent [5,6,7]. The molar tooth sign is usually considered pathognomonic and is related to the hypoplasia of cerebellar vermis and brain stem anomalies in axial plan on MRI [8,9,10]. More than 40 distinct genetic forms have been described, all following autosomal recessive inheritance (*AHI1*, *CC2D2A*, *CEP290*, *CPLANE1*, *TMEM67* being the most frequently involved), except for X-linked *OFD1* [2,11].

Autosomal recessive *KATNIP*-related JS was described for the first time in 2015 [12].

## 2. Materials and Methods

Here, we describe a male child with clinical features suggestive of a JS-related disorder, although he had esophageal atresia with tracheoesophageal fistula (EA/TEF) and a normal cerebral MRI without the molar tooth sign. Whole-exome sequencing (WES) identified a homozygous *KATNIP* pathogenic variant. WES was performed at R&I Genetics (Padova, Italy) on the NEXTSeq2000 platform (Illumina, San Diego, CA, USA) using the SureSelect All Exon V6 (Agilent, Santa Clara, CA, USA). Data were analyzed with BWA-GATK and proprietary pipelines, filtering for 1485 genes involved in neurodevelopmental disorders. Because of the result, the analysis was then extended to verify that there were no other significant variations.

The proband is the third child of consanguineous parents (Figure 1), who are first cousins of Roma ethnic background. Fetal ultrasounds were not performed regularly during pregnancy because of poor parental compliance. At birth, he showed type C esophageal atresia, which was surgically corrected on the second day of life but necessitated subsequent esophageal dilatation because of recurrence. Further congenital anomalies were bilateral post-axial polydactyly of the hands and an atrial septal defect, which resolved spontaneously. Rectal biopsies were performed for chronic constipation and possible Hirschsprung’s disease, but results were inconclusive. From the fifth month of life, he showed few episodes of flexion spasms and tonic-clonic seizures and was on anti-epileptic therapy for two years (no further episodes were reported). Cerebral MRI was performed, but no major parenchymal anomalies were detected except for slight diffuse enlargement of ventricular and periencephalic spaces. He had severe iron deficiency anemia, which required transfusion and was not responsive to martial supplementation.

He showed severe developmental delay, with absent language. At three years of age, he was 90 cm tall (3° centile), OFC was 47 cm (3° centile), weight 12.7 kg (3–10° centile), facial features were characterized by thick lips, cupped ears, anteverted nares, wide and sparse eyebrows (Figure 1). Ambulation was possible at four years, but only with support. At 5 years of age, a new clinical evaluation was performed: speech was absent, the same facial features were noted, he had a stature of 97 cm (<3° centile), weight was 15 kg (3° centile), and OFC was 48 cm (10–25° centile).

## 3. Results

Blood karyotype and Array-CGH were required as first tests, but no significant copy number variants were detected. Furthermore, IRIDA syndrome was suspected, based on unresponsive iron deficiency, but sequencing analysis of TMPRSS6 did not detect any pathogenic variants.

Afterwards, Whole Exome Sequencing was performed and filtered for genes associated with neurodevelopmental disorders. A c.808del (p.Ser270ValfsTer28) pathogenic variant in *KIAA0556* (*KATNIP*, NM_015202.3) was identified in a homozygous state in the proband, whereas both parents were heterozygous. No other pathogenic/likely pathogenic variants were reported. Segregation analysis of the *KATNIP* variant was extended to the older healthy brother and sister, and it was absent in both (Figure 1). We suggested ENT examination with polysomnography and an endocrinology evaluation for the short stature, but the parents refused the evaluations. The MRI was reevaluated, but no signs suggestive of JS could be observed (Figure 1). Ophthalmological evaluation could not be completed due to a lack of compliance.

## 4. Discussion

*KATNIP* pathogenic variants represent a rare cause of JS (Joubert Syndrome 26 OMIM #616784) [12,13,14,15,16,17]. *KATNIP* protein binds directly to cytoplasmic microtubules and increases their stabilization; in C. elegans, disruption of the Kiaa0556 ortholog resulted in normal ciliary structure, function, and transport, although significant microtubule defects were detected by ultrastructural analysis [12]. To the best of our knowledge, 11 individuals (eight males, three females) from seven independent families are currently reported (Table 1).

Enrichment of consanguineous marriages is expected for ultrarare recessive disorders and parents were consanguineous in 5/7 *KATNIP* families. Data should be taken with caution from such a small cohort, but *KATNIP*–related JS seems neurologically milder than other forms of JS. Three of 11 individuals nevertheless had a severe developmental delay: in at least two of these families, the clinical phenotype is blended due to the presence of a 2nd variant [14,16] and expected to be more complex due to the additive effect of distinct variants. One additional variant is a homozygous loss-of-function variant in *ADGRG1* [14], which is associated with complex brain malformations in humans [18]. In a second family [16], an additional variant of uncertain significance was reported in *KIF7*, which is a ciliopathy gene associated with variable phenotypes (including Joubert-12) [19] and is a strong candidate for disease modification. The third individual with a severe neurological phenotype is the proband of the current report, but we did not identify additional single-nucleotide or copy number variants involving genes that could explain a modification/worsening of the phenotype. *KATNIP*–related JS shows further peculiarities. Polydactyly, which is frequent in JS and a classical “ciliopathy” feature, was never reported before in individuals with *KATNIP* variants. Pituitary involvement is infrequent in other forms of JS but is clearly emerging as a specific feature of *KATNIP*-related JS: pituitary defects were described on brain MRI in at least 4/11, and endocrine abnormalities are common (three probands with proven growth hormone deficiency, one also had central hypothyroidism). Most individuals had short stature. Kidney abnormalities, which are frequent in JS, were never reported, but genital anomalies were common in males: 3/8 had cryptorchidism and/or micropenis, possibly due to pituitary hormone deficiency. On brain imaging, abnormalities of the corpus callosum and cerebellar hypoplasia were present (4/11 and 10/11, respectively), but four of 11 individuals did not show typical MTS. Furthermore, the proband of the current report showed severe unexplained anemia and EA/TEF. Iron deficiency anemia is often a multifactorial disorder, and the complex malformative/functional digestive issues of this child may be contributing factors; a hypothesis of IRIDA Syndrome was not confirmed by sequencing analysis of *TMPRSS6*. EA/TEF is a rare birth defect. It can be isolated, but it occurs within the context of additional anomalies in approximately 50% of individuals. It is a common feature of the VACTERL association spectrum, for which a specific molecular etiology is not known. Less than 10% of syndromic EA/TEF are caused by de novo chromosomal aberrations [20], and a few monogenic syndromes are known as well (e.g., Feingold syndrome caused by *MYCN* pathogenic variants) [21]. A possible involvement of ciliopathies in EA development is suggested by animal models, specifically, esophageal and tracheal anomalies were observed in a ciliopathy-associated mouse model with an Ift172 hypomorphic variant [22]. Esophageal atresia is not a specific feature of JS, but it is very interesting to note that *KATNIP* was already reported as a candidate gene for the etiopathogenesis of this defect [23], although one single individual was reported, with a de novo missense variant and a poor clinical description (esophageal atresia, heart defect, cleft lip). Pituitary anomalies are common and may lead to genital abnormalities in males and short stature. MTS can be absent, making the clinical diagnosis of JS challenging. The neurological phenotype is often mild but can be worsened by additional genetic variants in other genes that could affect clinical variability, as in the case of the patients described by Niceta and Cauley [14,16]. Thus, it is possible that further pathogenic variants, undetected due to technical limitations (e.g., small copy number variants or deep intronic variants), could modulate/influence the phenotype of this proband. Furthermore, the background genetic architecture could be studied through the implementation of polygenic risk scores and may act as a modifier of monogenic disorders [24]. Genome sequencing, which is now moving into clinical practice, would enable the analysis of intronic and regulatory regions and may soon help to determine the prognosis of rare disorders with better precision.

## 5. Conclusions

We describe the clinical features of a new proband with a homozygous loss-of-function *KATNIP* variant and compare these findings with the phenotype of a few previously reported individuals. Considering that our proband showed severe developmental delay and other rare features such as anemia unresponsive to iron supplementation and EA/TEF, it is intriguing to speculate about an undetected pathogenic variant (Deep intronic? Small copy number?) or an unfavorable genetic background. Nevertheless, we provide further evidence that ciliopathies contribute to esophageal atresia and identify *KATNIP* variants as an EA/TEF possible predisposing factor.

## Figures and Tables

**Figure 1 genes-16-00524-f001:**
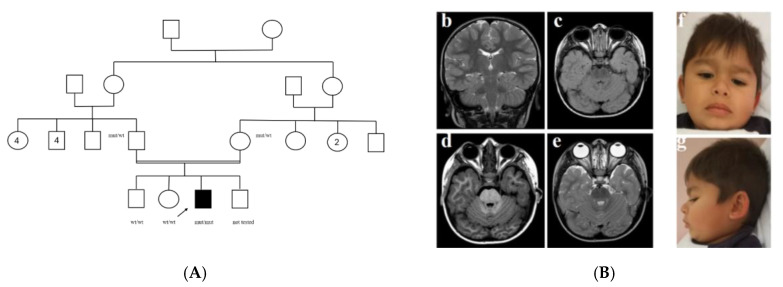
(**A**) Family tree and segregation of *KATNIP* variant. Central panel: (**B**) sagittal and (**b**–**e**) axial MRIs showing absence of molar tooth sign or other major abnormalities. Right panel: facial features at 5 years of age (**f**,**g**), showing wide and sparse eyebrows with synophrys, epicanthus, anteverted nares, thick lips. Parents signed an informed consent for publication of clinical pictures and results of genetic tests.

**Table 1 genes-16-00524-t001:** NR Not Reported; NA Not Applicable.

Phenotype and Gene Variants	Present Report	Aksu Uzunhan [13]	Niceta [16]	Cauley [14] (Two Brothers)	Fujita [15]	Roosing [17] (Two Siblings)	Sanders [12] (Three Siblings)
*KATNIP* Genotype	c.808del, p.(Ser270Valfs * 28) Homozygous	c.1461G>A, p.(Trp847 *) and c.4035delC, p.(Ile1346fs)	c.3756dupC, p.(Arg1253Glnfs * 5) Homozigous	c.222_232del, p.(Asn74GlufsTer11) Homozigous	c.2373del, p.(Asp791Glufs * 206) and c.4551+1G>A, p.(?)	c.4420del, p.(Met1474Cysfs * 11) Homozigous	c.2674C>T, p.(Gln892 *) Homozigous
Reported variants in other genes	No	No	*KIF7*: c.2675G>A, p.(Arg892His) Homozigous	*ADGRG1*: c.886C>T, p.(Gln296Ter) Homozigous	No	No	No
Gender	Male	Male	Male	Males	Female	Males	Two females, one male
Developmental Delay/Intellectual Disability	Yes, severe	No (2 years old)	Yes, severe	Yes, severe	Yes	Yes, mild	Yes, mild
Brain Abnormalities	Mild dilatation of periencephalic spaces	Arnold Chiari type I, ectopic neurohypophysis, mild cerebellar vermis hypoplasia	MTS, dysgenesis of the corpus callosum, ectopic posterior, and hypoplastic anterior pituitary	Mild MTS, bilateral polymicrogyria, hydrocephalus, diffuse white matter alterations, thin corpus callosum	MTS, agenesis of the corpus callosum, pituitary hypoplasia, hypothalamic hamartoma	MTS (both), thin corpus callosum (one)	MTS and hypoplastic pituitary in one, cerebellar vermis hypoplasia in two
Seizures	Flexion spasms and tonic-clonic seizures, no EEG abnormalities	No	No	Tonic-clonic seizures	Gelastic and tonic seizures	No	Occasional convulsions despite normal EEG recordings in one
Muscular Tone	Hypotonia	Normal	Hypotonia	Hypoyonia, generalized muscle wasting, later upper and lower limb spasticity, and hyperreflexia	Neonatal hypotonia	Mild hypotonia	Neonatal hypotonia in two
Facial Features	Thick lips, cupped ears, anteverted nares, wide and sparse eyebrows	Frontal bossing, sparse and broad-arched eyebrows, hypertelorism	Sparse hair, hypertelorism, thick eyebrow, bilateral ptosis, short columella, low-set ears	NR	NR	NR	Ptosis, frontal bossing, hypertelorism, anteverted nares
Growth Retardation	Yes	Yes	Yes	Yes	NR	NR	Yes
Eye Abnormalities	NA	No	Left microphthalmia and coloboma, right oculomotor nerve palsy, strabismus	Left-sided ptosis and ophthalmoplegia	Oculomotor apraxia	Oculomotor apraxia, nystagmus, and bilateral ptosis, cone dystrophy	Nystagmus
Other birth defects	Esophageal atresia, post-axial polydactyly, heart defect	Cryptorchidism	Micropenis, cryptorchidism, hand and feet brachydactyly, cleft lip and palate	NR	NR	No	One with micropenis and cleft lip and palate
Other features	Constipation, anemia	Combined pituitary hormone deficiency	GH deficiency, flat feet	Mild pes cavus and talipes (in one)	NR	No	Neonatal transient tachypnea, recurrent respiratory tract infections. Central hypothyroidism and GH deficiency

* indicates truncating mutation.

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
