# Peer review of "Phenotypic Spectrum of KATNIP-Associated Joubert Syndrome: Possible Association with Esophageal Atresia and Review of the Literature"

_genes, 2025, doi:10.3390/genes16050524_

Round 1

Reviewer 1 Report

Comments and Suggestions for Authors The authors identified a novel homozygous frameshift variant in the KATNIP gene in a 5-year-old boy with a syndromic phenotype using exome sequencing, describing his clinical features like severe developmental delay and esophageal atresia, despite the absence of the typical molar tooth sign (MTS) associated with Joubert syndrome (JS). The authors also reviewed the literature on KATNIP-related JS cases, compiling phenotypes from six previously reported families to highlight common features like pituitary abnormalities and distinct differences such as the lack of MTS and polydactyly in some cases.

This reviewer believe the manuscript should be accepted for publication after the following minor revisions:

1. the exact method for whole exome sequencing should be provided (kit/sequencing protocol? sequencing data analysis method?) Also, data availability (public depository? available upon request?), even not publicly accessible due to regulatory issues, should still be discussed.   2. The authors claimed in "5. conclusion", that "The neurological phenotype is often mild but can be worsened by additional genetic variants." Please provide further justification for this claim. The readers may be able to infer from the two cases in Table 1 but this should be clearly stated and discussed. 

Author Response

Comment 1: the exact method for whole exome sequencing should be provided (kit/sequencing protocol? sequencing data analysis method?) Also, data availability (public depository? available upon request?), even not publicly accessible due to regulatory issues, should still be discussed.

1:WES was performed at R&I Genetics (Padova, IT) on NEXTSeq2000 platform (illumina, San Diego, CA) using the SureSelect All Exon V6 (Agilent, Santa Clara,CA). Data were analyzed with BWA-GATK and proprietary pipelines, filtering for 1485 gene involved in neurodevelopmental disorders. Because of the result, the analysis was then extended to verify that there were no other significant variations.

Comment 2:The authors claimed in "5. conclusion", that "The neurological phenotype is often mild but can be worsened by additional genetic variants." Please provide further justification for this claim. The readers may be able to infer from the two cases in Table 1 but this should be clearly stated and discussed. 

2:The neurological phenotype is often mild but can be worsened by additional genetic variants in other genes that could affect clinical variability, as in the case of the patients described by Niceta and Cauley. Considering that our proband showed severe developmental delay and other rare features such as anemia unresponsive to iron supplementation and EA/TEF, it is intriguing to speculate about an undetected pathogenic variant (deep intronic? Small copy number?) or an unfavorable genetic background.

Reviewer 2 Report

Comments and Suggestions for Authors

Tedesco et al. report a case of KATNIP-associated Joubert syndrome, a rare genetic disorder. Only six families have been previously reported. The authors identified a c.808del, p.Ser270ValfsTer28 variant in KATNIP in a 5-year-old male from a consanguineous family of Roma ethnic background. The proband presented with severe developmental delay, hypotonia, postaxial polydactyly, severe anemia, and esophageal atresia. However, he did not exhibit the characteristic brain abnormality known as the molar tooth sign (MTS).

The manuscript is well-structured and scientifically sound. The figures are of good quality, and the tables are clear.

Major concern:

The main issue is novelty. I am not convinced the manuscript provides new insights. The phenotypic features are already known.

Minor comments:

  1. I would suggest submitting this as a case report rather than a review.

  2. The methodology of exome sequencing should be described in detail.

  3. The phenotypic description is too brief and does not belong in the Materials and Methods section.

  4. The Results section is missing.

  5. Consider replacing "Deambulation" with "ambulation" if that was the intended term.

  6. Instead of the section titled Figures, Tables, and Schemes, it should be renamed to Results.

  7. The Conclusion is too long; most of it should be moved to the Discussion section.

  8. English editing and proofreading would benefit the manuscript (e.g., gene names should be written in italics).

  9. Please confirm that parental consent was obtained for publishing the photo of the patient.

Author Response

Comments1:I would suggest submitting this as a case report rather than a review.

Response1:We indicated our article as review because we made a comparison between our patient and others reported with alterations of KATNIP gene.

Comment2:The methodology of exome sequencing should be described in detail

Response 2: We have inserted the methodology used

Comment 3:The phenotypic description is too brief and does not belong in the Materials and Methods section.

Response 3: We indicated main characteristics of the patient for the purpose of article.

Comment 4: The Results section is missing

Response 4:We changed the structure

Comment 5:Consider replacing "Deambulation" with "ambulation" if that was the intended term

Response 5:We changed with "ambulation".

Comment6: Instead of the section titled Figures, Tables, and Schemes, it should be renamed to Results.

Response 6: We changed the section

Comment7:The Conclusion is too long; most of it should be moved to the Discussion section.

Response 7: We changed it

Comment 8:English editing and proofreading would benefit the manuscript (e.g., gene names should be written in italics).

Response8: We have made the change

Comment 9:Please confirm that parental consent was obtained for publishing the photo of the patient.

Response 9: We had uploaded consent to publication

Round 2

Reviewer 2 Report

Comments and Suggestions for Authors

Thanks for your responses. The quality of the mansucript is good, th emain issue is originality. 

Comments:

"it is intriguing to speculate about an undetected pathogenic variant (deep intronic? Small copy number?) or an unfavorable genetic background"-this would fit better into disussion. Also, please explain what is meant by unfavorable genetic background

There are still some typos e.g. Homozigous in the table. 

Author Response

Comment 1

it is intriguing to speculate about an undetected pathogenic variant (deep intronic? Small copy number?) or an unfavorable genetic background"-this would fit better into disussion. Also, please explain what is meant by unfavorable genetic background

There are still some typos e.g. Homozigous in the table. 

Response 1 We expanded the discussion and added a reference to support.